# Seed Priming with Spermine Mitigates Chromium Stress in Rice by Modifying the Ion Homeostasis, Cellular Ultrastructure and Phytohormones Balance

**DOI:** 10.3390/antiox11091704

**Published:** 2022-08-30

**Authors:** Farwa Basit, Javaid Akhter Bhat, Zaid Ulhassan, Muhammad Noman, Biying Zhao, Weijun Zhou, Prashant Kaushik, Ajaz Ahmad, Parvaiz Ahmad, Yajing Guan

**Affiliations:** 1Institute of Crop Science, Zhejiang Key Laboratory of Crop Germplasm, Zhejiang University, Hangzhou 310058, China; 2Hainan Research Institute, Zhejiang University, Sanya 572025, China; 3International Genome Center, Jiangsu University, Zhenjiang 212013, China; 4Independent Researcher, 46022 Valencia, Spain; 5Department of Clinical Pharmacy, College of Pharmacy, King Saud University, Riyadh 11451, Saudi Arabia; 6Department of Botany, GDC, Pulwama 192301, Jammu and Kashmir 192301, India

**Keywords:** seed priming, spermine, chromium, reactive oxygen species, phytohormones

## Abstract

Chromium (Cr) is an important environmental constraint effecting crop productivity. Spermine (SPM) is a polyamine compound regulating plant responses to abiotic stresses. However, SPM-mediated tolerance mechanisms against Cr stress are less commonly explored in plants. Thus, current research was conducted to explore the protective mechanisms of SPM (0.01 mM) against Cr (100 µM) toxicity in two rice cultivars, CY927 (sensitive) and YLY689 (tolerant) at the seedling stage. Our results revealed that, alone, Cr exposure significantly reduced seed germination, biomass and photosynthetic related parameters, caused nutrient and hormonal imbalance, desynchronized antioxidant enzymes, and triggered oxidative damage by over-accretion of reactive oxygen species (ROS), malondialdehyde (MDA) and electrolyte leakage in both rice varieties, with greater impairments in CY927 than YLY689. However, seed priming with SPM notably improved or reversed the above-mentioned parameters, especially in YLY689. Besides, SPM stimulated the stress-responsive genes of endogenous phytohormones, especially salicylic acid (SA), as confirmed by the pronounced transcript levels of SA-related genes (OsPR1, OsPR2 and OsNPR1). Our findings specified that SPM enhanced rice tolerance against Cr toxicity via decreasing accumulation of Cr and markers of oxidative damage (H_2_O_2_, O_2_^•−^ and MDA), improving antioxidant defense enzymes, photosynthetic apparatus, nutrients and phytohormone balance.

## 1. Introduction

Rice (*Oryza sativa*) is the second most abundant cereal crop and fulfils the nutritional needs of at least 50% of the global population [1]. Soil contamination with environmental pollutants (mainly of anthropogenic origin) such as heavy metals (HMs) has caused severe rice yield losses in Asian countries, including China [2]. Rice plants taking up HMs from contaminated soils and their depositing in plant parts, mainly grains, leads to reduction in crop yields [3]. Chromium (Cr) is the seventh most hazardous metal, which usually exists in Cr^+3^ and Cr^+6^ forms, while Cr^+6^ imposes more lethal symptoms on plants, relative to Cr^+3^ [4]. After taking up Cr ions from soils, plants generally accumulate an excessive amount of Cr in their tissues, severely hindering their growth and development [5,6,7,8]. At morpho-physiological, biochemical, metabolic and cellular levels, Cr inhibits the rate of seed germination, plant growth indices (length/height and biomass), photosynthesis process, and accumulation of mineral nutrients [1,7], induces oxidative stress, viz., lipid peroxidation, and reactive oxidative species (ROS), damages the inside of cellular membranes and ultra-structures [8,9], and provokes disorganization in antioxidant enzyme activities [10]. Plants boost their internal antioxidant defense system, which includes enzymatic and non-enzymatic antioxidants, by minimizing the overproduction of ROS and the induced oxidative cellular damage [7,11,12,13].

Polyamines (PAs) are aliphatic compounds and osmo-protectants, with low molecular weight, located within plant cells, and control diverse plant functioning such as regulation of seed germination, embryogenesis, flower expansion, fruit development and ripening, against both normal and stress conditions [6,14]. Moreover, PAs improve the plant’s tolerance capacity by enhancing the endogenous accumulation of PAs under various environmental stresses, i.e., water-deficit [15], salt [16], chilling [17], heat [18] and heavy metals [19]. Previous findings have revealed that exogenous applications of PAs improved plants’ growth indices by improving levels of chlorophyll pigments, photosystem II, the antioxidant enzymatic defense system and membrane protection, and reducing the HM-induced cellular oxidative damage, as noticed under cadmium [20], lead [21] and chromium [22] stress. Furthermore, PAs such as spermine (SPM) contribute to regulation of cellular functioning, cell division and phytohormonal signal transductions in cadmium and copper exposed wheat leaves [23]. The key mechanism is that the exogenous supply of PAs stimulates the endogenous PA content, which may help plants boost their intrinsic immunity in response to outside environmental stressors, improving the morpho-physiological and biochemical attributes of mung bean [20], wheat [24], rice [25], and maize [22]. However, the contribution of spermine when used as a priming agent in the detoxification of Cr in rice plants has been less commonly investigated. Therefore, the current study was performed to obtain insights into the protective roles of SPM in the alleviation of Cr stress in sensitive (CY927) and tolerant (YLY689) rice cultivars by targeting plant growth traits, photosynthetic apparatus (chlorophyll pigments and photosystem II), nutrients uptake, Cr-accumulation, oxidative damage, antioxidant enzyme activities, endogenous phytohormone production, and membrane or cellular ultra-structures. These biomarker studies may help plant scientists fully understand the mechanistic approaches utilized by SPM to detoxify Cr in rice, and possibly other cereals grown in HM-polluted soils.

## 2. Materials and Methods

### 2.1. Availability of Seeds

Herein, two various rice varieties, Chunyou 927 (CY927, sensitive) and Yliangyou 689 (YLY689, tolerant), obtained through the Zhejiang Nongke Seeds Co., Ltd. Hangzhou, Zhejiang Province, China were used in the present study.

### 2.2. Seed Priming and Germination Analysis

Initially, rice seeds were dipped into the solution for sterilization with 5% (*w*/*v*) of NaClO for 20 min, then quickly splashed with purified water (ddH_2_O) to eradicate the remaining chloride. Secondly, these sterilized seeds were further primed by 0.01 mM SPM and water (H_2_O) separately at 30 °C in the dark for 24 h. Then, to restore the seed’s original moisture levels, they were dried at ambient temperature. The seeds primed with water (H_2_O) were taken as controls. After priming, germination of seed assessment was conducted by using 50 seeds for each box (12 cm × 18 cm) with three repetitions for each treatment. Formerly, all germination boxes were retained inside a growth incubator with a fluctuating phase of 8 h light and 16 h of darkness at 25 °C for 14 days [26]. The chromium (0 and 100 µM) was applied to the incubated seeds. The Cr concentration was selected based on the preliminary experiments. The total seeds germinated, counted on the fifth day, were considered as the germination energy (GE) [26]. Furthermore, the total germinated seeds counted on day 14 were used for calculation of percentage of germination (GP), germination index (GI), vigor index (VI), and mean germination time (MGT), using the formulas used by Zheng [26].

### 2.3. Hydroponic Culture Treatments and Plant Growth Analysis

Subsequently, the primed seeds were placed into a 96 well black hydroponic box with one seed in each well. Each treatment was repeated six times and each repetition had 80 seeds. The hydroponic nutrient solution consisted of 0.5 µM potassium nitrate, 0.5 µM Ca(NO_3_)_2_, 0.5 µM magnesium sulfate, 2.5 µM monopotassium phosphate, 2.5 µM ammonium chloride, 100 µM ferric EDTA, 30 µM boric acid, 5 µM manganese sulfate, 1 µM copper sulfate, 1 µM zinc sulfate, and 1 µM ammonium hepta-molybdate ((NH_4_)_6_Mo_7_O_24_). The hydroponic boxes were repositioned daily within the growth chamber (30 °C with an alternation cycle of 12/12 h light/dark) by completely randomized design (CRD). After two weeks of hydroponics, three repeats of each treatment were exposed with 100 µM Cr for 7 days, as well as another three repeats without Cr, used as control. Sampling was carried out after the 7th day of Cr treatment.

The sampling of 21-day-old seedlings was conducted and they were washed with ddH_2_O to exterminate the Cr residues. The height of seedlings was calculated by using a ruler and their fresh biomass was measured on a scale. To estimate the dry mass, leaves, and roots were dried out in an oven separately at 75 °C for 24 h.

### 2.4. Determination of Photosynthetic Pigments

The estimation of chlorophyll contents such as chlorophyll a (*Chl* a), chlorophyll b (*Chl* b), total chlorophyll as *Chl* (a + b) and carotenoids (Car) was carried out following the method of Lichtenthaler and Wellburn [27]. Concisely, fresh leaves tissues (0.2 g) were homogenized with pure water and soaked in 3 mL of 95 percent ethanol (*v*/*v*). The mixture was centrifuged for 10 min at 5000× *g* to separate the supernatant. After that, supernatant at the volume of 1 mL aliquot were mixed with 9 mL ethanol (95 percent, *v*/*v*). Thereafter, using an exhausting spectrophotometer, the mixture was measured using absorbance at 665, 649, and 470 nm wavelengths [28]. The following equation was used to determine the chlorophyll contents:Chlorophyll a (*Chl* a) = 13.95 A665 − 6.88 A649(1)
Chlorophyll b (*C**hl* b) = 24.96 A649 − 7.32 A665(2)
Total chlorophyll content = *Chl* a + *Chl* b(3)
Carotenoids (C_x+c_) = (1000 A470 − 2.05 C_a_ − 104 C_b_)/245(4)

The values of photosynthetic attributes were measured as milligrams (mg) per liter (L) of plant extract. Parameters related to gaseous exchange were investigated by the methodology of Zhou and Leul [29]. After 2 h of assimilation inside the growth cabinet at 18 °C, 1000 molm^−2^ s^−1^ light intensity, and 60% comparative moisture, the uppermost wholly extended leaves were used to calculate the rate of transpiration (Tr), net photosynthetic rate (Pn), stomatal conductance (gs), the intercellular level of CO_2_ (Ci), and photochemical efficacy of PS II (*Fv*/*Fm*). ImagingWin software (IMAGING-PAM, Walz, Effeltrich, Germany) was used to examine the colored images for *Fv*/*Fm* and *Fm* levels.

### 2.5. Estimation of Cr Contents

The samples of dried root and shoot (0.2 g) per treatment were mixed with 5 mL concentrated HNO_3_ and HClO_4_ (5:1, *v*/*v*) on a stovetop at 70 °C for around 5 h. Dilution of the samples (digested) was subjected to dilution to a final amount of 10 mL with 2% HNO_3_ before being tested, and three replicates per treatment were prepared. To examine the Cr along with microelements Na, Cu, K, P, Fe, Ca, Mn, and Zn, scum was measured by an atomic absorption spectrometer (iCAT-6000-6300, Thermo Scientific, Waltham, WA, USA).

### 2.6. Determination of Electrolyte Leakage and Total Soluble Sugar

The rice seedling is utilized to estimate electrolyte leakage (dSm^−1^). Surface sterilization was carried out for 5 g seeds with HgCl_2_ (1%) and speckled with ddH_2_O, with four replications. Subsequently, seedlings were drenched in 25 mL ddH_2_O, and were left at room temperature for one-day in an incubator. The sample was moved to a new void beaker, up to the volume of 25 mL by adding ddH_2_O. Electrolyte leakage was reported in dSm^−1^ [30]. To estimate the total soluble sugar, samples of 0.5 g in the form of fresh shoots were crushed with pestle and mortar, in an extraction buffer that prepared from phosphate (50 mM, pH 7), ascorbate (1 mM), KCl (100 mM), glycerol (10%, *v*/*v*) and β-mercapto-ethanol (5 mM). Then, the supernatant was amassed into the micro-centrifuge tube through centrifugation at 12,000× *g* for 15 min. Later, the homogenate of three replicates was used to quantify the total soluble protein content [31] and total soluble sugar, by the procedure as for the phenol-sulfuric acid assay [32].

### 2.7. Estimation of Endogenous Abscisic Acid, Jasmonic Acid and Salicylic Acid Contents

The quantification of endogenous ABA contents was accomplished by using frozen samples following the protocol of Kim et al. [33] and Kamboj et al. [34]. The endogenous contents of JA were quantified following the detailed procedure of [35]. Extraction and quantification of free SA was carried out via the protocol of Yalpani et al. [36] and Fang et al. [37]. Three biological replicates were used to estimate the endogenous abscisic acid, jasmonic acid and salicylic acid contents.

### 2.8. Estimating MDA, H_2_O_2_ and O_2_^•−^ Contents

The determination of MDA content was performed with 2-thiobarbituric acid (TBA). Homogenization of ~2 mL extract in 3 mL of TBA (5%) was performed, before being diluted inside 5% of trichloroacetic acid (TCA). Formerly, the grounded and mixed samples were preheated for 15 min at 95 °C, and earlier cooled to ice instantaneously and centrifuged at two different wavelengths, viz., 532 nm and 600 nm, via utilizing a UV–vis spectrophotometer (Hitachi U-2910) [38]. The value of MDA content was donated as nmol mg-1 protein. The hydrogen peroxide (H_2_O_2_) was determined by following the detailed protocol of Kwasniewski [39]. The H_2_O_2_ contents were estimated as µmol g^−1^ FW. Superoxide radical (O_2_^•−^) contents were estimated following the method of Jiang and Zhang [40] with a few amendments. Three replications were used to estimate the MDA, H_2_O_2_ and O_2_^•−^ contents.

### 2.9. Determination of Antioxidant Enzyme Activities

The supernatant attained from total soluble sugar was supplementarily used to determine the activities of antioxidants. Three replications were used to determine these antioxidant activities. To observe the SOD activity, the protocol utilized by Giannopolitis and Ries [41] was pursued with minor modifications. In addition, the SOD activity was estimated as U g^−1^ FW. Activity of catalase was estimated following the approach of Aebi [42]. The peroxidase (POD) activity was assessed by following Change and Maehly [43] detailed protocol. The enzyme activity (APX, POD, CAT) was deliberated as μ mol min^−1^ mg^−1^ protein at 25 ± 2 °C [44].

### 2.10. Analysis of Gene Expression

The transcript levels of SA-related genes were estimated by qRT-PCR. In a mortar and pestle, thawed shoot samples of both rice cultivars were crushed inside liquid nitrogen. Trizol method was used to extract the RNA, as already defined [45]. The NanoDrop 2000/2000 c (Thermo-Fisher Scientific, Waltham, MA, USA) was used to identify the purity of RNA. Subsequently, the synthesis of cDNA was accomplished by PrimeScript™ RT reagent kit. Three replications were used to analyze the gene expression level. The *PR1*, *PR2*, and *NPR1* gene primers were utilized to analyze the concerned gene expressions. The used primers are listed in Appendix A. The 2XSYBR Green Master Mix reagent (10 µL volume), (Applied Biosystems, Foster City, CA, USA), 200 nM gene-specific primers, and cDNA samples (6 µL volume) were utilized to prepare the 20 µL reaction mixture. The relative alteration inside the expression of genes was identified as documented [46]. The house-keeping gene (*OsActin*) was utilized as a control gene to standardize the other genes during internal calibration.

### 2.11. Transmission Electron Microscopic Analysis

The leaf ultrastructure changes were observed by the Sheteiwy et al. [47] procedure with minute modifications. The leaf samples without veins (7-10 per treatments) were randomly selected after treatment applications, then immersed in 2.5 percent (*v*/*v*) glutaraldehyde in 0.1 M PBS (sodium phosphate buffer, pH 7.4) and eroded thrice with the same PBS. Moreover, the 1% OsO_4_ (osmium (VIII) oxide) was used to postfix the leaves for nearly 1 h. Further, it was washed three times in 0.1 M PBS with a 15 min gap between each wash. In addition, leaf samples were dehydrated using various classified categories of ethanol, such as 50%, 60%, 70%, 80%, 90%, 95%, and 100%, correspondingly, and washed by using absolute acetone for 20 min. Later, samples were immersed overnight in Spurr’s resin. Thus, ultra-thin segments (80 nm) of samples were amended, then retained inside copper nets for visualization via transmission electron microscope (JEOLTEM-1230EX) at 60.0 kV.

### 2.12. Statistical Analysis

The experimental results were investigated by applying analysis of variance (One-way ANOVA) with the least significant differences (LSD) at *p* < 0.05 and 0.01 levels between mean values using Statistix (8.1) software. Three replications were used for each experiment and standard errors (S.E) were represented in figures.

## 3. Results

### 3.1. Priming Effect of SPM on Plant Phenotype, Seed Germination and Plant Growth Traits under Cr Exposure

In this study, seeds priming with water were utilized as a control. Under control conditions, plant height was visibly enhanced with SPM treatment as compared to seed primed control plants (Appendix A). The Cr treatments alone significantly decreased the plant height/length and yellowing/burning of leaves compared to their controls, while severe toxic symptoms were observed in rice cultivar CY927 than YLY689. Under Cr stress, seed priming with SPM reversed the Cr toxicity as noticed by the improved overall plant height/length and greenish texture of leaves (Appendix A). In control treatments, no significant difference in the GE, PG, GI and VI of rice seeds were observed. Nevertheless, the Cr-exposure alone caused a significant reduction in seed germination parameters such as GE, GP, GI, and VI relative to controls (Table 1). This decrease was more pronounced in CY927 as compared to YLY689 under Cr-alone treatments. The mean germination time (MGT) was noticeably increased in CY927 compared to YLY689 under Cr applications, whereas seed priming with SPM significantly reduced the MGT under Cr stress. In contrast, seeds primed with SPM noticeably enhanced the GE, GP, GI and VI of rice seedlings under Cr exposure (Table 1). It was observed that these germinations indices were more improved in YLY689 when compared to CY927.

Concerning the morphological changes, Cr-stress alone significantly reduced the plant height including shoot length (S/L), and root length (R/L), along with plant biomass including fresh weight (F/W) and dry weight (D/W). Compared with seeds primed with water (H_2_O) in the nutrient only solution, seeds primed with SPM enhanced plant growth as well as biomass, especially in YLY689 under Cr stress (Table 1). These outcomes designated that seed priming with SPM mitigated the toxic effects of Cr on rice seedlings and improved the seed germination, plant growth and biomass efficiently under Cr-stress conditions, with more pronounced improvement noticed in YLY689 than CY927.

### 3.2. Priming Effect of SPM on Photosynthetic Pigments, Photosystem II and Gas Exchange Parameters under Cr Stress

Compared to control plants, seeds primed with SPM treatments marginally enhanced photosynthetic pigments such as chlorophyll-a (*Chl* a), chlorophyll-b (*Chl* b), chlorophyll-(a + b) (*Chl* (a + b)) and carotenoids in both rice cultivars, while treatments with Cr alone significantly reduced the *Chl* a (68.6% and 34.4%), *Chl* b (47.5% and 29.13%), *Chl* (a + b) (43.5% and 26.2%), and carotenoids (40.9% and 29.2%) in CY927 and YLY689, respectively (Figure 1). Seed priming with SPM enhanced the *Chl* a, *Chl* b, *Chl* (a + b), and carotenoids by 30.1%, 28.4%, 25.2%, 33.6% in CY927, and in 36.3%, 31.7%, 29.6%, 38.9% YLY689, respectively, under Cr stress (Figure 1a–d). Current investigations displayed that seed priming with SPM increased the *Chl* a, *Chl* b, *Chl* (a + b) and carotenoids in both rice cultivars as compared to respective controls in both treatments, with and without Cr toxicity. Likewise, the SPM application enhanced the *Fv*/*Fm* values as compared to relative controls, and the increment was 11.4% more in YLY689 than CY927 under Cr exposure (Appendix A). In addition, this was further verified by taking visual images of *Fv*/*Fm*, and *Fm*. The rice leaves were exposed to 100 µM Cr stress more than untreated plants, showing as light blue/green, correspondingly, by decreasing the *Fv*/*Fm*, and *Fm* ratio (Figure 1e–h).

The exposure to Cr alone notably declined the values of gas exchange parameters including Pn, Tr, Gs, and Ci by 57.1%, 46.8%, 32.0%, 49.3% in CY927, and 38.7%, 33.1%, 23.9%, and 42.11% in YLY689 rice cultivars, correspondingly, but more decline was noticed in the values of the CY927 cultivar than YLY689 (Appendix A). Nevertheless, seeds primed with SPM significantly improved the values of gas exchange indices under Cr toxicity in both CY927 and YLY689 rice cultivars. In control conditions, non-significant differences in the values of Pn, Tr, gs, Ci were noted between seeds primed with water and SPM of both cultivars. These findings revealed that seeds primed with SPM mitigated the toxic effects of Cr on photosynthetic pigments and gas exchange parameters in both cultivars, but this effect was more prominent in YLY689 relative to the CY927 cultivar.

### 3.3. Priming Effect of SPM on Electrolyte Leakage, Total Soluble Sugar and Total Soluble Protein under Cr Stress

In control conditions (seed priming with water and SPM without Cr toxicity), no noticeable changes in the values of electrolyte leakage (EL), total soluble sugar (TSS) and total soluble protein (TSP) were observed in the leaves of both rice cultivars (Figure 2a–c). However, Cr stress alone caused a prominent increase in the values of EL (78.4% in CY927, and 43.7% in YLY689) (Figure 2a) but decreased the TSS and TSP levels (49.2/36.5% in CY927, and 37.4/31.2% in YLY689, respectively) (Figure 2b,c), when compared to the control without Cr stress. The increasing or decreasing trends of EL, TSS and TSP values were more pronounced in CY927 (in the case of EL) than YLY689 (in the case of TSS and TSP), although results represented that seed priming with SPM reduced the EL (32.9% and 45.7%), and increased the TSS and TSP by 24.7/19.3% in CY927 and 32.2/21.6% in YLY689 cultivar, respectively, when compared to their relative controls under Cr stress (Figure 2a–c). These outcomes indicated that SPM notably minimized the EL but improved the TSS and TSP levels compared to seed priming with water, especially in YLY689 under Cr stress. These outcomes suggested that SPM notably reversed the Cr-mediated induction in EL and reduction in TSS, as well as TSP in rice seedlings.

### 3.4. Priming Effect of SPM on Hydrogen Peroxide (H_2_O_2_), Superoxide (O_2_^•−^) and Malondialdehyde (MDA) under Cr Stress

Under Cr alone treatments, the accumulation of H_2_O_2_ (91.2/72.5% and 67.1/55.8%), O_2_^•−^ (76.9%/67.1% and 51.8/47.3%), and MDA (77.6/74.2% and 59.2/46.9%) were significantly induced in shoots/roots of CY927 and YLY689, correspondingly, as compared to control plants. Inclusively, a high accumulation of H_2_O_2_, O_2_^•−^ and MDA was noticed in shoots rather than roots of both cultivars (Figure 2d–i). The accretion of H_2_O_2_, O_2_^•−^ and MDA contents were more prominent in CY927 than in YLY689, though seed priming with SPM markedly decreased the accumulation of H_2_O_2_ (38.8/27.3% and 33.4/24.9%), O_2_^•−^ (35.2/26.3% and 24.9/18.3%) and MDA (47.8/37.1% and 38.2/23.7%) in shoots/roots of both varieties (CY927 and YLY689), correspondingly, under Cr stress (Figure 2d–i). In control treatments, no significant difference was noticed among the H_2_O_2_, O_2_^•−^ and MDA content values in the roots and shoots of both rice cultivars.

To verify the accumulation of H_2_O_2_ and O_2_^•−^ inside the shoots and roots of both cultivars of rice, leaves were stained with DAB and NBT in response to seed priming with water, SPM and Cr treatments (Figure 3a–h). Compared to untreated control, leaves treated with Cr only revealed dark brown as well as dark blue staining, correspondingly, for H_2_O_2_ and O_2_^•−^. CY927 exhibited more dark staining colors than YLY689 indicating that CY927 accumulates more H_2_O_2_ or O_2_^•−^ than YLY689. In contrast, control treatments (seed primed with water and SPM without Cr addition) displayed a slight staining intensity of DAB and NBT in both cultivars (Figure 3a–h). These differences revealed that seed priming with SPM sustained the plasma membrane integrity and reduced the oxidative damages against Cr-induced overproduction of ROS in both rice cultivars (particularly in YLY689) in comparison to seed priming with water against Cr toxicity conditions.

### 3.5. Priming Effect of SPM on Cr Contents and Mineral Nutrients under Cr Stress

Under Cr alone treatments, the Cr accumulation was more pronounced in roots than shoots of both rice cultivars in seed priming with water under Cr stress. The Cr augmentation was reduced significantly in both shoots and roots in seed primed with SPM under Cr exposure. Micro- as well as macro-nutrients’ (Na, Mg, K, P, Ca, Mn, Fe, Cu, and Zn) uptake and translocation imbalance were observed under Cr toxicity compared to their controls. The decrease was more noticeable in roots than shoots of both CY927 and YLY689 cultivars (Appendix A). The uptake of Cr was greater in YLY689 as compared to the CY927. Under Cr exposure, seeds primed with SPM remarkably enhanced the Mn, Zn, Cu, P, Fe, and K in both roots and shoots of both cultivars. Moreover, SPM treatment modulated the uptake of Na inside both roots and shoots of rice cultivars (Appendix A). These outcomes revealed that seed priming with SPM restricted the Cr uptake, accumulation and translocation inside rice plants’ parts and further improved the macro- and micro-nutrient balance mandatory for plant development.

### 3.6. Priming Effect of SPM on Antioxidative Enzyme Activities under Cr Stress

Under Cr alone treatments, an upsurge in the activities of antioxidant (SOD, CAT, POD, and APX) activities were noticed inside the shoots and roots of both cultivars, but this increment was more conspicuous in shoots compared to roots. The outcomes demonstrated, compared to control plants, the Cr toxicity significantly surged the enzymatic activities i.e., SOD (58.1/47.9, 62.9/54.7%), APX (58.9/52.2, 65.9/63.1%), POD (70.4/58.5, 82.1/67.3%), and CAT (56.2/48.4, 64.9/53.2%) in shoots/roots of both cultivars CY927, and YLY689, respectively (Figure 4a–h). Interestingly, seed primed with SPM further enhanced the enzymatic activities such as SOD (29.1/16.2%, 38.7/19.5%), APX (35.7/26.6, 41.1/34.3%), and POD activity by 23.9/22.4, 28.6/25.9%, and CAT activity by 39.5/24.7%, 44.4/32.9% inside shoots/roots of both cultivars CY927, and YLY689, correspondingly (Figure 4a–h). Our outcomes revealed that seeds primed with SPM developed a higher possibility of scavenging the reactive oxygen species and increased plant tolerance capability with greater effect in YLY689 than CY927.

### 3.7. Priming Effect of SPM on the Production of Endogenous ABA, JA and Free SA Levels under Cr Stress

The ABA, JA and SA levels were examined in seeds primed with water or SPM, with and without Cr stress. The exposure of Cr alone led to elevated contents of ABA (2.83 and 2.49-fold), JA (3.32 and 3.29-fold) and SA (2.48 and 1.89-fold) in both varieties CY927 and YLY689, individually, when compared to controls (Figure 5a–c). Under Cr stress, seed priming with SPM applications significantly decreased the ABA level (Figure 5a), whereas no significant difference was observed in the levels of JA under Cr treatments (Figure 5b). Nevertheless, the level of SA was enhanced significantly (2.23 and 1.76-fold) in both rice cultivars (CY927 and YLY689), individually, by the applications of seed primed SPM under Cr stress (Figure 5c). A similar pattern was seen in the seedlings of both rice cultivars, with YLY689 displaying more obvious effects than CY927.

### 3.8. Priming Effect of SPM on Hormone-Associated Gene Expression Analysis under Cr Stress

Under Cr stress, the transcript levels of *PR1*, and *PR2* genes were subsequently upregulated in both CY927 and YLY689 compared to relative controls. Furthermore, the transcription levels of both these genes were markedly stimulated within seeds primed by SPM against Cr-induced stress as compared to the treatments of Cr alone in both rice cultivars (Figure 5d,e). The expression levels were noticeably upregulated in CY927 as compared to YLY689 (*p* < 0.01). Likewise, the upregulation in *NPR1* levels was noticed under Cr stress in both rice cultivars. Nevertheless, their expression levels were further increased in the seeds treated with SPM priming under Cr exposure (Figure 5f). An identical drift was noted in both rice varieties. The upregulation was more pronounced in the expression of the *NPR1* gene as compared to *PR1*, and *PR2* genes. This indicated that *NPR1* has a dynamic role in stimulating the phytohormone synthesis and mitigation of Cr toxicity. In addition, gene expression analysis supported the notion that SPM mediated modifications in SA contents under Cr applications. Our findings clearly represented that SPM has extensive participation in rice tolerance against Cr stress and modulates the levels of transcription of certain stress-responsive genes associated with phytohormones.

### 3.9. Priming Effect of SPM on Cellular Ultrastructural Changes under Cr Stress

Within control group (seeds primed with water and SPM without Cr stress), the leaf ultrastructure of both cultivars CY927 and YLY689 displayed a well-shaped cell wall, vigorous chloroplast, mitochondria, vacuoles, peroxisomes and the usual organized granule thylakoids inside grana containing normal granum thylakoids, stroma thylakoids, starch grains and plastoglobuli (Figure 6A,B,E,F). However, the ultrastructural analysis of rice cultivar (CY92, seeds primed with water) compared with Cr toxicity indicated a ruptured nuclear membrane, expansion of double-layered nuclear membrane, swollen mitochondria and damaged chloroplast (Figure 6C). Relatively, seeds primed with SPM presented a developed nuclear membrane, normal structure of chloroplast, plastoglobuli, and less swollen mitochondria than seeds primed with water beneath Cr exposure in CY927 (Figure 6D). In YLY689, seeds priming water (with Cr disclosure) revealed structural abnormalities such as thylakoids disruption, slightly irregular structured starch grains, mitochondrial damages, and swollen chloroplast (Figure 6G) than their relative control groups (Figure 6E,F). Interestingly, the seeds primed with SPM under Cr treatments minimized the structural damages induced by Cr stress in the mesophyll cells of cultivar YLY689 as observed by the well-developed cell wall, peroxisomes, vacuoles, well-shaped stroma thylakoids, starch granules, chloroplast, granum thylakoids, plastoglobuli and matured mitochondria (Figure 6H).

## 4. Discussion

Soil contamination with chromium (Cr) has become a serious concern for rice researchers due to its direct entry into food chains and severe health issues globally [48]. Thus, our prime focus was to investigate latent approaches to diminish Cr-accumulation and its associated phytotoxicity. For this purpose, we utilized seed priming with spermine (SPM) in minimizing the Cr-tempted toxic effects by targeting the seed germination indices, growth and biomass, photosynthetic apparatus, soluble protein levels, nutrients balance, oxidative damages, antioxidant enzymes’ defense systems, phytohormone production and modifications in cellular ultrastructure. Our findings showed that the seed germination parameters, i.e., GE, GP, VI, GI and MGT, were significantly reduced by Cr exposure in both rice cultivars (Table 1) which ultimately minimized the plant development, biomass (fresh and dry weight, shoot or root lengths) (Table 1, Appendix A). Comparatively, seed primed SPM caused readjustments of seed germination indices, plant growth and biomass attributes under Cr toxicity (Table 1 and Appendix A). These alterations were more substantial in YLY689 (tolerant variety) than CY927 (sensitive variety). Recent studies revealed that Cr-stress inhibited the rates of seed germination indices [1]. This may be due to the alteration in seed coat structure and reactions with ions, electrons or intact radicals. Consequently, sugar supply to the seeds was compromised [49,50]; although SPM applications aided the conversion of starch hydrolysis into soluble sugar that improved the respiration of seeds and starch metabolism, and consequently enhanced plant growth and biomass production [51].

Furthermore, Cr also inhibited the biomass of rice seedling [1,11], maize [22], and *Brassica napus* [7] seedlings that might be associated with a disturbance of photosynthetic systems and nutrients imbalance. As predicted, Cr stress significantly reduced the chlorophyll pigments (Figure 1a–d), *Fv*/*Fm* and *Fm* values (Figure 1e–h), gas exchange parameters (Appendix A) and nutrient accumulation (Appendix A) in both rice varieties, especially CY927. The possible reason is that Cr caused inhibition inside the production of chlorophyll biosynthetic enzymes (δ-amino-levulinic acid dehydratase and protochlorophyllide reductase), which led to the impairment of chlorophyll biosynthetic pathways [52], while SPM priming mitigated Cr stress on both cultivars, and precisely in YLY689. Potentially, seed primed SPM boosted the plant growth traits (Table 1) by eliminating Cr accumulation via maintaining the nutrient uptake balance (Appendix A), cell cycle and cell division, which led to the improvement in chlorophyll pigments (Figure 1a–d) and gas exchange indices (Appendix A) under Cr exposure. Most probably, SPM application enhanced the levels of photosynthetic apparatus by regulating the nutrient uptake which might have modulated the expression of genes associated with metal homeostasis, lessening the Cr accretion under heavy metal stress [53,54]. It has been reported that photosynthetic efficiency maintains the level of total soluble sugar (TSS) and total soluble protein (TTP) that might be stimulating the metabolites’ induction, and further helps to detoxify the Cr stress under Cr-induced toxicity [55,56]. Our upshots specified that the TSS and TSP are directly associated with the photosynthetic contents (Figure 2b,c) and the rates of gas exchange indices (Appendix A). Seed priming with SPM might have improved the photosynthetic and biosynthetic enzyme activities (i.e., δ-aminolaevulinic acid and protochlorophyllide reductase) that regulated the photosynthetic biosynthesis pathways [57] and subsequently improved the TSS and TSP. Our results revealed that seed primed SPM minimized the Cr-indulged oxidative damages and minimized the electrolyte leakage.

In our current investigations, Cr exposure enhanced the extra accumulation of H_2_O_2_ and O_2_^•−^ (Figure 2f–i), MDA contents (Figure 2d,e), electrolyte leakage (Figure 2a), particularly in CY927, and caused oxidative or membrane damages and lipid peroxidation as noticed in rice seedlings [57,58], Indian mustard [58] and tomato [59]. Seeds primed with SPM may have improved the ROS hemostasis by decreasing over-production of ROS (Figure 2d–i) that revealed the role of SPM in mitigating oxidative damages and subsequently improving photosynthetic performance. Probably, SPM improved the catabolism, maintained the cellular metabolism, and inhibited the overproduction of free radicals which controlled the level of MDA contents, electrolyte leakage, and ROS over-generation. Additionally, seed primed SPM treatment provided stability to cellular membranes by reducing the MDA contents (Figure 2d,e) and electrolyte leakage (Figure 2a) in rice seedlings under Cr-induced stress, thus enhancing the photosynthetic apparatus, plant growth and biomass production, specifically in YLY689. SPM capability in reducing ROS (H_2_O_2_ and O_2_^•−^) over production was verified by histochemical straining, which matches recent observations under conditions of metal stress [21,60].

To cope with the Cr-triggered oxidative damage, plants stimulate their antioxidative defense mechanisms as indicated by the upregulation in enzymatic activities such as SOD, APX, POD and CAT in both rice cultivars (Figure 4a–h). Similar outcomes were documented in a recent investigation that plants increase their antioxidative defense mechanism to scavenge Cr-induced cellular oxidative damage [1]. Seed priming with SPM further stimulated the antioxidative enzyme activities and reduced the ROS or MDA accumulation, which revealed that SPM boosted the antioxidants’ defense mechanisms and reduced the oxidative damage, thus maintaining ROS homeostasis in rice seedlings. Previous studies documented that PAs regulate antioxidant activities to eliminate the ROS extra generation, and hence maintain the redox balance, and thus improve the growth traits of wheat [21], *Brassica juncea* [61], and mung bean [20] under heavy metals stress. Expectedly, the sole treatment of Cr significantly restricted the accumulation of nutrients (Na, K, P, Ca, Mn, Zn, Cu and Fe) which eventually leads to lowered biomass production. The excessive Cr accumulation in plant parts may limit the accessibility of nutrients and disturb their hemostasis [1,7]. SPM applications displayed the capably to improve the balance of nutrients by lowering the Cr accretion via a vacuole sequestering approach and enhanced the tolerance capacity of YLY689 against Cr stress than CY927. Likewise, earlier investigations supported our results that the reduction in Cr-facilitation can improve the nutrients’ bioavailability within the tissues of *Raphanus sativus* [62] and maize [22].

It has been reported that phytohormones, viz., SA and JA interactions with PAs including SPM and spermidine (SPD), modulated plant physiological attributes, and enhanced antioxidant enzyme activities in scavenging the extra accumulation of ROS under adverse environmental stresses [63,64,65,66]. Under heavy metal stress conditions, SA levels were enhanced inside plant parts that helped to reduce the extra accumulation of ROS [67,68]. Remarkably, seeds primed with SPM significantly improved the endogenous levels of SA rather than JA under Cr toxicity in both rice varieties, especially in YLY689 as compared to CY927 (Figure 5b,c). Hereafter, we targeted the gene expression of SA-related genes (*PR1*, *PR2*, and *NPR1*) to analyze the genetic responses under Cr stress (Figure 5d–f). We verified that Cr significantly stimulated the endogenous SA levels that clarified the potential roles of SA in alleviating Cr toxicity in rice seedlings. Nevertheless, Cr displayed no significant differences in the levels of endogenous JA (Figure 5b). This might be due to the fact that Cr stress, in addition to the consequent synthesis of JA, stimulated the lipases to deliver a substrate for lipoxygenase [33,69]. Still, the molecular basis of this phenomenon needed thorough exploration.

Notably, Cr-alone treatments enhanced the endogenous abscisic acid (ABA) levels, while seed priming with SPM significantly lowered them (Figure 5a). Conceivably, PAs may regulate the ABA levels to modulate the signal transduction pathways, and specific protein’s activation and repression, which supplementarily controls the cell growth or cell death [70]. Therefore, PAs interact with phytohormones to maintain the plant’s growth attributes, stomatal conductance, and chlorophyll contents to alleviate the nastiest effects of multiple toxicities [66].

The ultrastructural analysis revealed that Cr alone treatments severely damaged the cellular organelles (Figure 6A–H) and CY927 demonstrated more damaging effects than YLY689. The extra accumulation of Cr within cell organs ruptured the chloroplast structure by disturbing the granule thylakoid development, swollen mitochondria and damages to thylakoid membrane, as supported by earlier studies [71]. The Cr-induced chloroplast damage led to the reduction in chlorophyll contents which halted the photosynthetic rate and gas exchange parameters [71]. Conversely, seed primed SPM applications reestablished the structures of granule thylakoids, stroma, chloroplasts, thylakoid membrane, cell wall, vacuoles and mitochondria under Cr stress, specifically in YLY689 (Figure 6d,h). Possibly, SPM repaired the cellular ultrastructure (chloroplast and thylakoids) that helps to reduce the overproduction of ROS and improve the photosynthetic apparatus, thus leading to higher plant development and biomass against heavy metals’ exposure [72].

## 5. Conclusions

A recent study reported that seeds primed with SPM exhibited a great capability to mitigate Cr-induced adverse effects in rice seedlings of both cultivars. Cr (100 µM) disclosure in nutrients media severely damaged the plant biomass/growth, impaired the photosynthetic pigment, imbalanced the micro- and macro-nutrients, augmented the cellular oxidative damage, altered the endogenous phytohormone level and cellular ultrastructure, and desynchronized the antioxidant defense system in rice seedlings, especially in CY927. Simultaneously, seeds primed with SPM ameliorated biomass/growth production, chlorophyll pigments, total soluble sugar, total soluble protein, maintained nutrient balance, and improved antioxidants’ defense system by lowering the Cr uptake, accumulation, translocation, lipid peroxidation, oxidative damage, and electrolyte leakage. Moreover, SPM modulated the endogenous phytohormones content, as well as the transcription level of SA-related genes (*PR1*, *PR2*, and *NPR1*) under Cr stress. Henceforth, it was revealed that SPM can enhance plant tolerance and mitigate the Cr phytotoxicity in different rice varieties. Furthermore, Cr stress damaged the rice cultivar CY927 more than YLY689. YLY689 was found more tolerant to Cr stress and SPM further enhanced tolerance capacity, while CY927 displayed susceptibility to Cr stress whose negative effects were reversed by SPM.

## Figures and Tables

**Figure 1 antioxidants-11-01704-f001:**
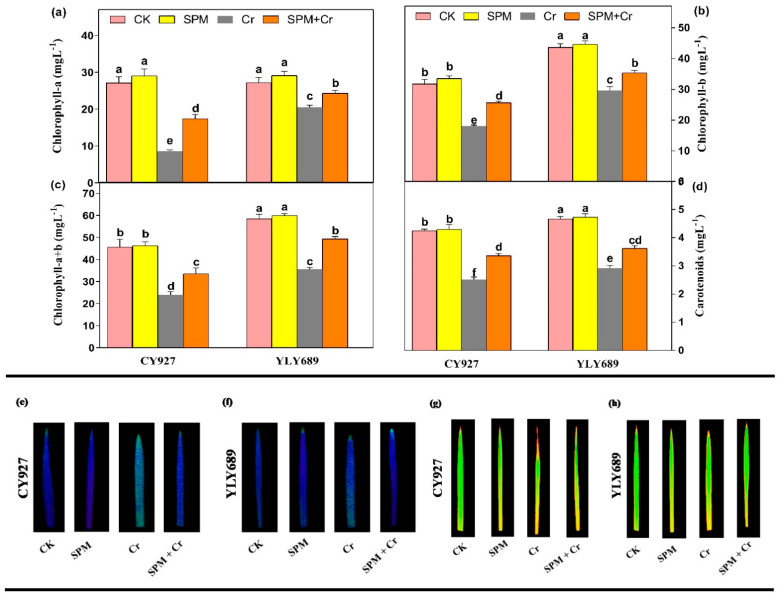
Effects of seed primed SPM on photosynthetic pigments (**a**) chlorophyll–a, (**b**) chlorophyll–b, (**c**) total chlorophyll–(a + b), and (**d**) carotenoids, (**e**,**f**) *Fv*/*Fm* levels in the leaves of two rice cultivars CY927 and YLY689, respectively and (**g**,**h**) *Fm* levels in the leaves of two rice cultivars CY927 and YLY689, correspondingly under chromium (Cr) stress. Values are mean ± SE of three independent replicates and different letters (a–f) above bars show a significant difference between treatments by LSD test at *p* < 0.05.

**Figure 2 antioxidants-11-01704-f002:**
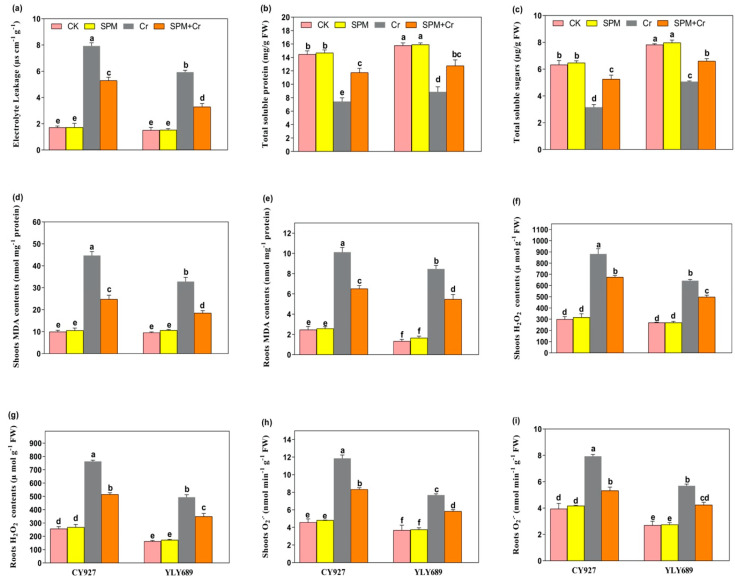
Effects of seed primed SPM on (**a**) electrolyte leakage (EL), (**b**) total soluble proteins, (**c**) total soluble sugar (TSP), (**d**) MDA contents in shoots, (**e**) MDA contents in roots, (**f**) H_2_O_2_, contents in shoots (**g**) H_2_O_2_, contents in roots (**h**) O_2_^•−^ contents in the shoots and (**i**) O_2_^•−^ contents in the roots of two rice varieties (CY927 and YLY689) against chromium (Cr) stress. Values are mean ± SE of three independent replicates and different letters (a–f) above bars show a significant difference between treatments by LSD test at *p* < 0.05.

**Figure 3 antioxidants-11-01704-f003:**
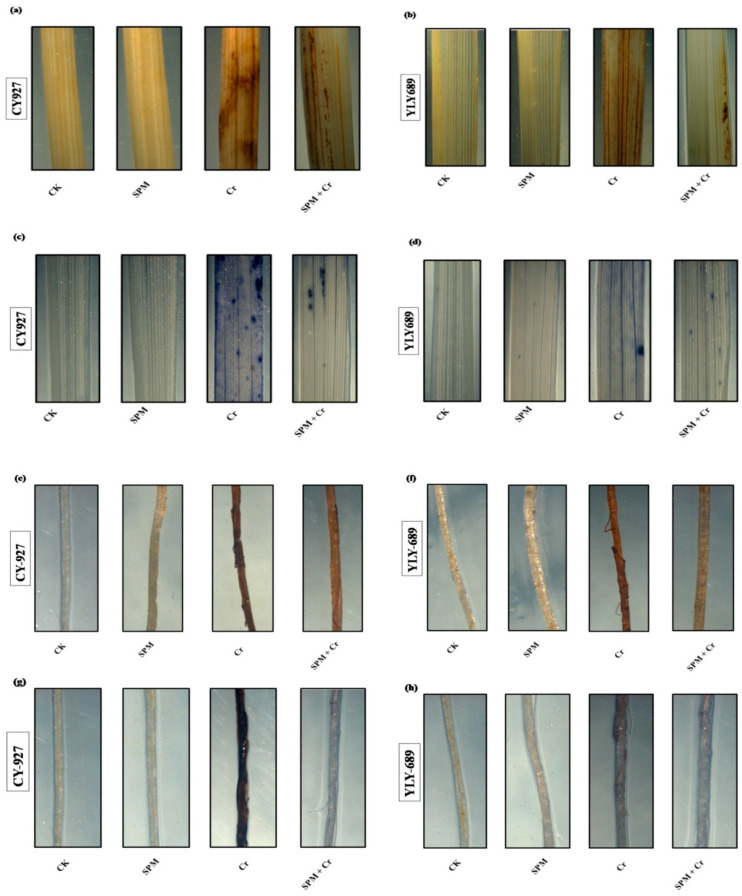
Effects of seed primed SPM on histochemical staining to identify the accumulation of (**a**) hydrogen peroxide (H_2_O_2_) in the shoots of CY927, and (**b**) hydrogen peroxide (H_2_O_2_) in the shoots of YLY689 by 3,3-diaminobenzidine (DAB), (**c**) superoxide (O_2_^•−^) in the shoots of CY927, and (**d**) superoxide (O_2_^•−^) in the shoots of YLY689 by nitro blue tetrazolium (NBT), (**e**) hydrogen peroxide (H_2_O_2_) in the roots of CY927, and (**f**) hydrogen peroxide (H_2_O_2_) in the roots of YLY689 by 3,3-diaminobenzidine (DAB), (**g**) superoxide (O_2_^•−^) in the roots of CY927, (**h**) superoxide (O_2_^•−^) in the roots of YLY689 by nitro blue tetrazolium (NBT).

**Figure 4 antioxidants-11-01704-f004:**
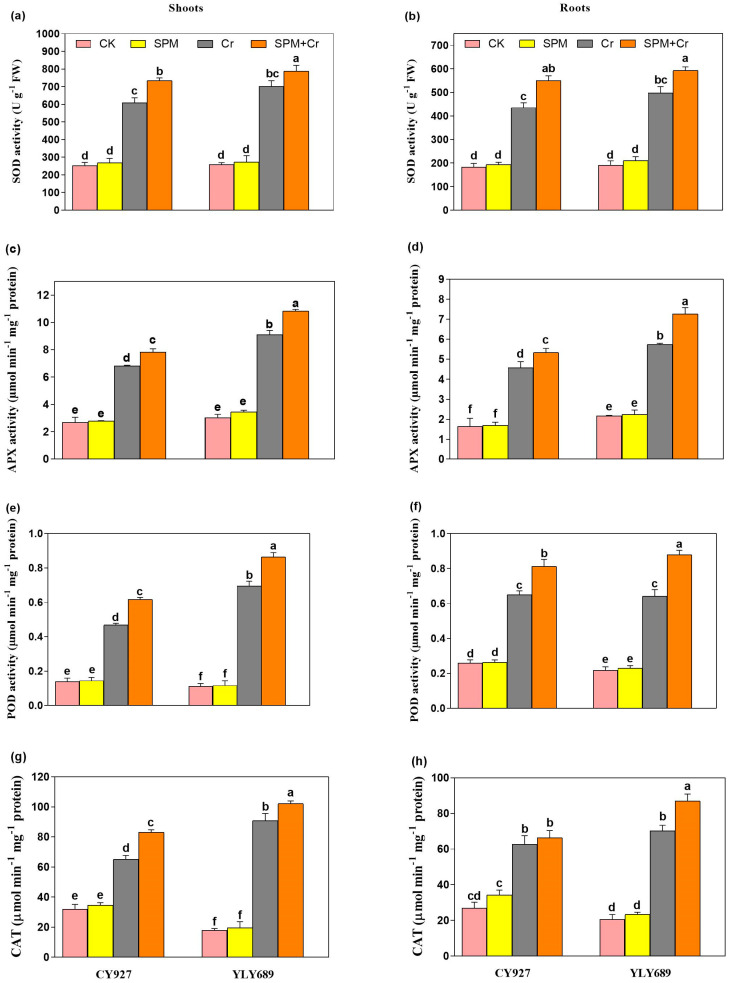
Effects of SPM on the activities of (**a**) superoxide dismutase (SOD) in shoots, (**b**) superoxide dismutase (SOD) in roots, (**c**) ascorbate peroxidase (APX) in shoots, (**d**) ascorbate peroxidase (APX) in roots, (**e**) peroxidase (POD) in shoots, (**f**) peroxidase (POD) in roots, (**g**) catalase (CAT) in shoots, and (**h**) catalase (CAT) in roots of both rice varieties (CY927 and YLY689) under chromium (Cr) stress. Values are mean ± SE of three independent replicates and different letters (a–f) above bars show a significant difference between treatments by LSD test at *p* < 0.05.

**Figure 5 antioxidants-11-01704-f005:**
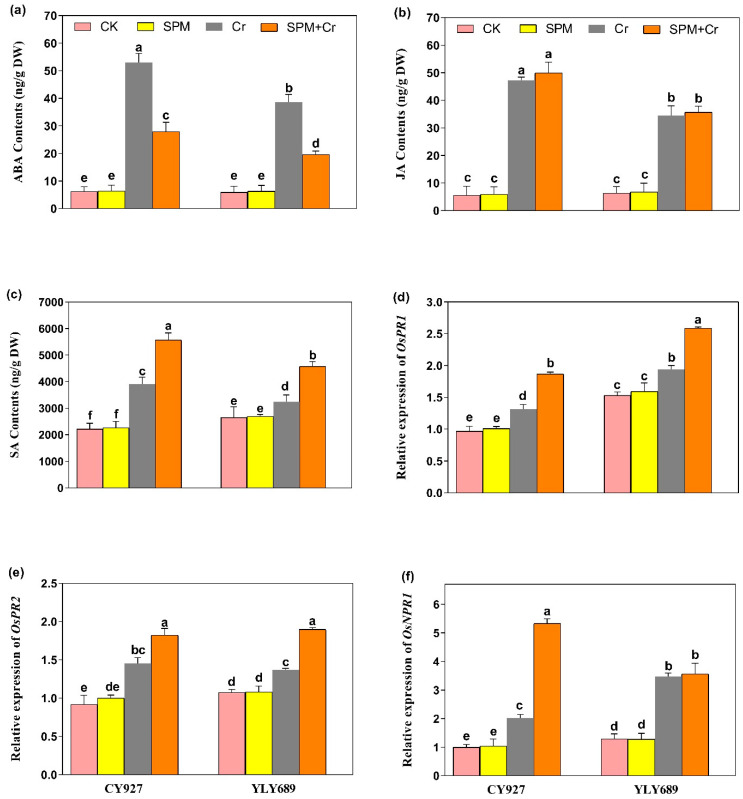
Effects of SPM on the endogenous production of phytohormones including (**a**) abscisic acid (ABA), (**b**) Jasmonic acid (JA), (**c**) salicylic acid (SA) and expressions of phytohormones-associated genes (**d**) *PR1*, (**e**) *PR2*, and (**f**) *NPR1* in two rice cultivars (CY927 and YLY689) under chromium (Cr) stress. Values are mean ± SE of three independent replicates and different letters (a–f) above bars show a significant difference between treatments by LSD test at *p* < 0.05.

**Figure 6 antioxidants-11-01704-f006:**
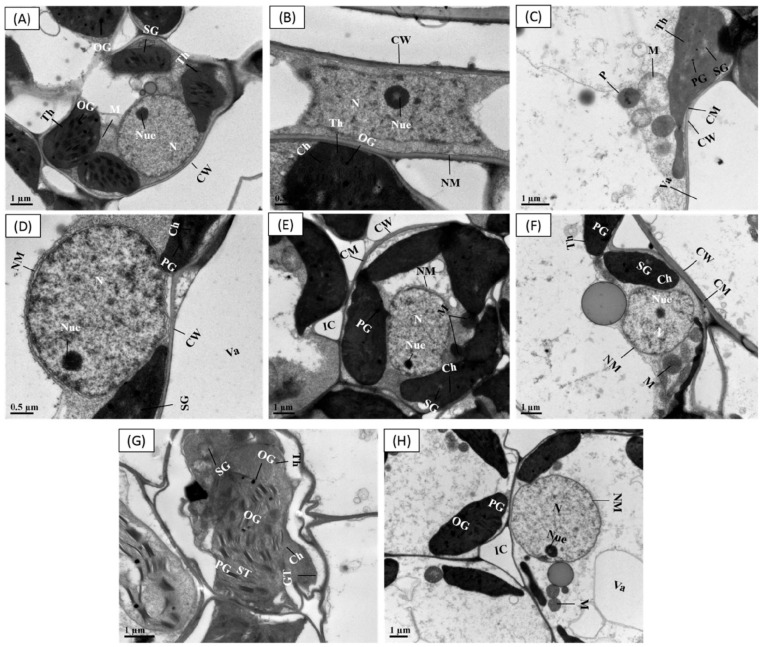
Effect of seeds primed with SPM on cellular ultra-structural changes in the leaves of two different rice cultivars (CY927 and YLY689) against chromium (Cr). (**A**) Leaf mesophyll cell of CY927 (primed with water) at control level, (**B**) Leaf mesophyll cell of CY927 (primed with Spm) at control level, (**C**) Leaf mesophyll cell of CY927 (primed with water) under Cr stress, (**D**) Leaf mesophyll cell of CY927 (primed with Spm) under Cr toxicity, (**E**) Leaf mesophyll cell of YLY689 (primed with water) at control level, (**F**) Leaf mesophyll cell of YLY689 (primed with Spm) at control level, (**G**) Leaf mesophyll cell of YLY689 (primed with water) under Cr stress, (**H**) Leaf mesophyll cell of YLY689 (primed with Spm) under Cr toxicity. Abbreviations: N (nucleus), SG (starch grain), NM (nuclear membrane), PG (plastogloboli), CW (cell wall), Va (vacuole), Ch (chloroplast), CM (cell membrane), (chloroplast), GT (granule thylakoids); M (mitochondria), ST (stomata thylakoids).

**Table 1 antioxidants-11-01704-t001:** Effects of seed priming with SPM on germination energy (GE), germination percentage (GP), germination index (GI), mean germination time (MGT), Fresh weight (F/W), dry weight (D/W), vigor Index, shoot length (S/L), and root length (R/L) of rice seeds under Cr toxicity.

Genotypes	Treatments	GE (%)	GP (%)	GI	MGT (days)	F/W (g)	D/W (g)	V.I	S/L (cm)	R/L (cm)
CY927	CK	92.00 ± 5.29 ^b^	98.67 ± 2.31 ^a^	22.26 ± 0.61 ^b^	2.70 ± 0.08 ^d^	1.05 ± 0.04 ^c^	0.10 ± 0.00 ^a^	2.27 ± 0.09 ^b^	16.47 ± 0.02 ^b^	13.89 ± 0.02 ^c^
	Spm	93.33 ± 1.15 ^ab^	100.00 ± 0.00 ^a^	23.36 ± 0.24 ^ab^	2.72 ± 0.04 ^d^	1.09 ± 0.02 ^b^	0.10 ± 0.00 ^a^	2.30 ± 0.01 ^b^	16.79 ± 0.06 ^b^	14.24 ± 0.06 ^b^
	Cr	34.67 ± 2.31 ^f^	35.33 ± 1.15 ^e^	6.59 ± 0.51 ^f^	3.59 ± 0.38 ^a^	0.56 ± 0.01 ^f^	0.06 ± 0.00 ^d^	0.36 ± 0.02 ^f^	8.25 ± 0.03 ^f^	5.27 ± 0.03 ^g^
	Spm + Cr	72.67 ± 1.15 ^d^	87.33 ± 2.31 ^c^	18.21 ± 0.96 ^d^	3.29 ± 0.06 ^b^	0.86 ± 0.01 ^d^	0.08 ± 0.00 ^c^	1.46 ± 0.04 ^d^	12.04 ± 0.02 ^e^	10.37 ± 0.02 ^e^
YLY689	CK	96.67 ± 3.06 ^a^	100.00 ± 0.00 ^a^	25.69 ± 0.73 ^a^	2.69 ± 0.10 ^d^	1.18 ± 0.01 ^a^	0.10 ± 0.00 ^a^	2.66 ± 0.09 ^a^	17.67 ± 0.02 ^ab^	15.36 ± 0.02 ^a^
	Spm	98.00 ± 2.00 ^a^	100.00 ± 0.00 ^a^	26.12 ± 0.46 ^a^	2.71 ± 0.06 ^d^	1.19 ± 0.02 ^a^	0.10 ± 0.01 ^a^	2.70 ± 0.10 ^a^	18.57 ± 0.07 ^a^	15.42 ± 0.07 ^a^
	Cr	67.33 ± 1.15 ^e^	69.33 ± 1.15 ^d^	15.30 ± 0.05 ^e^	3.47 ± 0.38 ^ab^	0.69 ± 0.01 ^e^	0.08 ± 0.00 ^c^	1.17 ± 0.03 ^e^	13.95 ± 0.02 ^d^	9.76 ± 0.02 ^f^
	Spm + Cr	79.33 ± 4.16 ^c^	90.00 ± 2.00 ^b^	20.43 ± 0.91 ^c^	3.01 ± 0.06 ^c^	0.88 ± 0.01 ^d^	0.09 ± 0.00 ^b^	1.80 ± 0.09 ^c^	15.02 ± 0.12 ^c^	12.21 ± 0.12 ^d^

The same letters within a column designate that there was no significant difference at a 95% probability level at the *p* < 0.05 level according to LSD test, correspondingly.

## Data Availability

The data are contained within the article and Appendix A.

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
