# Peer review of "Seed Priming with Spermine Mitigates Chromium Stress in Rice by Modifying the Ion Homeostasis, Cellular Ultrastructure and Phytohormones Balance"

_antioxidants, 2022, doi:10.3390/antiox11091704_

Round 1

Reviewer 1 Report

The manuscript include valuable results on plant protective mechanisms against Cr stress. Thus, it is appropriate for the publication. In the same time, the manuscript should be checked and improved.

Check affiliations of the authors.

Line 53. “Polyamines (PAs) are the aliphatic compounds or osmoprotectants…”. Replace “or” with “and”

Line 63. “Furthermore, PAs (SPM) contributes”. Add “spermine” before the abbreviation SPM.

Line 71. “in the alleviation of Cr” - alleviation of Cr stress?

Line 89. “The seeds primed water” - primed with water?

Line 124. “The following equation was exploited to determine the chlorophyll contents:” Include the equation.

Line 169. “the method of Jiang and Zhang, (2001)” – add reference.

Line 217 and throughout the text. Figure S1 (as well as Table S1, etc.) is not provided.

Page 8, Line 29 and throughout the text. Designations of the columns (a, b, d, c, bc, etc.) in Figures 1, 2, 4, and 5 are not clear.

Line 88 “Figure 3. Effects of seed primed SPM on the histochemical staining of shoots to identify the accu-88 mulation of hydrogen peroxide (H2O2) and superoxide (O2• ‒) by 3, 3-diaminobenzidine (DAB) and 89 nitro blue tetrazolium (NBT), correspondingly.” Include designations of the variants (a-h) in Figure 3 caption.

Line 176. Figure 6 caption. Include designations (A)-(H) in the caption.

Author Response

Please see the attachment, the response to reviewers 1 comments are present in it.

Reviewer 2 Report

Overall the paper is of interest and the experiment conducted well.  There are a number of issues with language and data presentation that are mentioned below.

Title

  1. Looks fine

Abstract

  1. Well written and highlighting the most important information

Introduction

  1. Good justification for the study.
  2. English grammar should be checked carefully through the manuscript.  There are a number of issues just within the introduction.

Material and Methods

  1. Line 94-95:  What are the primary experiments that determined Cr concentration?  Do you mean a preliminary investigation?
  2. Line 124 – equation is missing
  3. Line 119 – Was the leaf tissue homogenized before being soaked in the ethanol?
  4. Overall the experiment was well designed.
  5. Carefully check word choice throughout the methods.  Some of the word choices are distracting.  For example – Line 156 – ‘ensuring’ should be replaced with either ‘following’, or ‘combining’ depending on what was done.

Results and Discussion

  1. Is Germination Energy a standard term with a standard definition and if so please cite the reference.
  2. How did you decide which data was shown directly in the table and which would be supplementary data tables?
  3. Once again carefully check grammar and word usage.

Figures and tables

  1. In the tables be sure to define the abbreviations used in the table columns within the table legend.  Do the same in all other figure and table legends.
  2. In figure 1 be sure to indicate which parts of the figure refer to each of the measured quantities.  It is unclear what e-h are.  Same with all other figures.
  3. In addition all figure and table legends should include a reference to the type of statistical test that was conducted.

References

  1. For the in text references sometimes you say following the procedure of [number] and others you list the author and then the number.  Be sure to be consistent.

Author Response

Please see the attachment, authors response reviewers 2 comments.

Round 2

Reviewer 1 Report

Authors have improved the manuscript, by the following point should be clarified:

“Point 9. Line 217 and throughout the text. Figure S1 (as well as Table S1, etc.) is not provided.

Response 9: We agree with this assessment. Now, we have provided the Table S1, and Figure S1 in text. (Yellow color highlighted in the revised manuscript, Line no. 187-188, 216, 220).”

Supplementary tables and figures are referred in the text, but they are not included in the manuscript and are not provided as additional material for the article. They should be provided in some form. 

Author Response

Please see the attachment, response to all comments has been provided. Thanks
